# Ets transcription factor GABP controls T cell homeostasis and immunity

Chong T. Luo [1,2], Hatice U. Osmanbeyoglu[3], Mytrang H. Do[1], Michael R. Bivona[1], Ahmed Toure[1], Davina Kang[1], Yuchen Xie[2], Christina S. Leslie[3] & Ming O. Li[1]

Peripheral T cells are maintained in the absence of vigorous stimuli, and respond to antigenic stimulation by initiating cell cycle progression and functional differentiation. Here we show that depletion of the Ets family transcription factor GA-binding protein (GABP) in T cells impairs T-cell homeostasis. In addition, GABP is critically required for antigen-stimulated T-cell responses in vitro and in vivo. Transcriptome and genome-wide GABP-binding site analyses identify GABP direct targets encoding proteins involved in cellular redox balance and DNA replication, including the Mcm replicative helicases. These findings show that GABP has a nonredundant role in the control of T-cell homeostasis and immunity.

[1] Immunology Program, Memorial Sloan Kettering Cancer Center, New York, NY 10065, USA. [2] Louis V. Gerstner Jr. Graduate School of Biomedical Sciences, Memorial Sloan Kettering Cancer Center, New York, NY 10065, USA. [3] Computational Biology Program, Memorial Sloan Kettering Cancer Center, New York, NY 10065, USA. Correspondence and requests for materials should be addressed to M.O.L. (email: lim@mskcc.org)

The peripheral naive T-cell population is maintained in number, diversity, and functional competence under steady-state conditions[1]. This homeostasis relies on signals from T-cell receptor (TCR) self-peptide major histocompatibility complex interaction and the common gamma chain cytokine interleukin 7 (IL-7)[2]. Upon microbial challenge, pathogen-specific T cells grow in size, followed by robust proliferation and differentiation into effector T cells[3]. Disruption of naive T-cell homeostasis and effector T-cell responses results in debilitating and lethal diseases associated with immunodeficiency[4].

A multitude of transcription factors have been described as important regulators of T-cell responses. For example, the fork-head box O (Foxo) family of transcription factors are essential for naive T-cell survival and trafficking, in part through the regulation of IL-7 receptor α-chain (IL-7Rα), L-selectin (CD62L) and the chemokine receptor CCR7[5]. In addition, the E twenty-six (Ets) family of transcriptional factors, characterized by a conserved DNA-binding domain that recognizes nucleotide sequences with a GGAA/T core motif, have been implicated in T-cell regulation[6]. T cells deficient in Ets1 are more susceptible to cell death[7, 8]. By contrast, depletion of Elf4 results in enhanced homeostatic and antigen-drive proliferation of CD8+ T cells[9], suggesting that Ets proteins can function as both positive and negative modulators of peripheral T-cell responses.

Compared with other Ets family transcription factors, GA-binding protein (GABP) is a unique member as it functions as an obligate multimeric complex[10]. GABP is composed of GABPα, which binds to DNA through its Ets domain but lacks transactivation capability, and GABPβ that is recruited by GABPα and contains the transcription activation domain[11, 12]. GABPα has a single transcript isoform that is widely expressed across tissue types, whereas GABPβ has multiple isoforms and some can dimerize, allowing for the formation of a GABPα$_2$/β$_2$ hetero-tetramer complex[13, 14]. Targets of GABP include housekeeping genes, such as those involved in ribosomal and mitochondrial biogenesis[10, 15, 16], which might account for the embryonic lethal phenotype of GABPα-deficient mice[17, 18]. GABP also regulates tissue-restricted targets such as acetylcholine receptors in neuro-

muscular synapse and integrin-β2 in myeloid cells[19, 20]. Moreover, GABP has been shown to facilitate the progression of multiple cancers, including chronic myeloid leukemia, liver cancer, and glioblastomas[21–24].

Studies of GABP in T cells have mainly focused on its role in the control of Il7ra transcription[18]. Analysis of embryonic thymocytes from mice harboring constitutive depletion of the Gabpa gene revealed a complete abolishment of IL-7Rα expression[18]. A later report using Lck-Cre to trigger conditional knockout of Gabpa gene from CD4−CD8− double-negative (DN) 1-DN2 thymocytes showed that T-cell development was arrested at the DN3 stage[25]. However, IL-7Rα expression was not defective in DN3 thymocytes, and it was only partially reduced in DN4 cells[25]. Furthermore, ectopic expression of IL-7Rα failed to alleviate the DN3 block caused by GABPα ablation[25], suggesting that GABPα regulation of early T-cell development is independent of IL-7Rα. Nevertheless, it is unclear whether GABPα regulates IL-7Rα expression in mature T cells, and whether GABPα has additional functions in the control of T-cell homeostasis and effector T-cell responses.

In this report, we utilize a mouse model that ablates GABPα from CD4+CD8+ double-positive (DP) thymocytes. We find that although T-cell development is largely unperturbed, loss of GABPα triggers a diminishment of peripheral T-cell populations. In vitro culture experiments show that GABPα is crucial for T-cell activation, proliferation, and survival upon antigen challenge. Mechanistic studies identify GABP target genes involved in the control of cellular redox balance, DNA replication, and cell cycle progression. Consequently, depletion of GABPα impairs T-cell homeostatic survival, proliferation, and antigen-induced responses in vivo. Collectively, our findings identify GABP as a central regulator of T-cell homeostasis and T-cell immunity.

## Results

**T-cell development is unperturbed in $CD4^{Cre}Gabpa^{f/f}$ mice.** GABPα deficiency in T-cell progenitors results in compromised T-cell development[18, 25]. To study the function of GABPα beyond

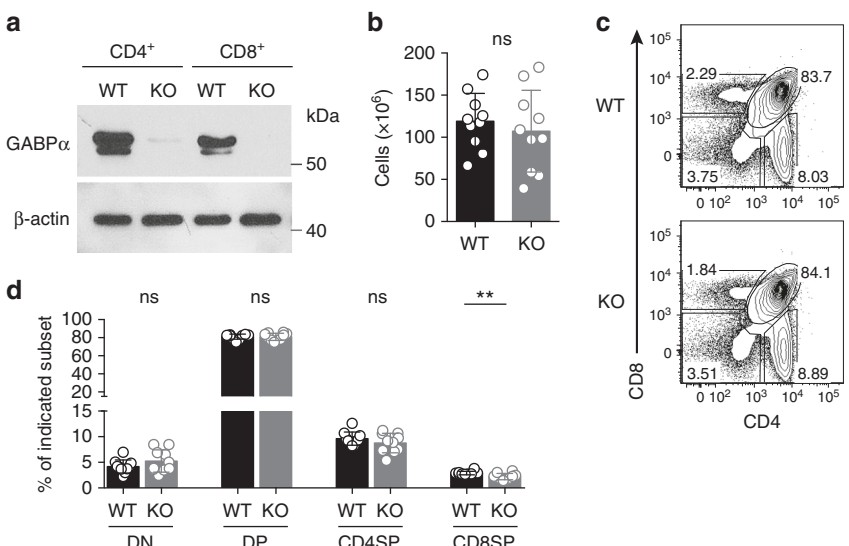

**Fig. 1** Thymocyte development in T-cell-specific GABPα-deficient mice. **a** Immunoblotting analysis of GABPα in purified CD4+ and CD8+ T cells from the spleens and lymph nodes (LNs) of Gabpa$^{f/f}$ (wild-type, WT) and CD4$^{Cre}$Gabpa$^{f/f}$ (KO) mice. β-actin was used as a loading control. Uncropped western blotting images were included in Supplementary Fig. 10. **b** Total numbers of thymocyte in WT and KO mice. **c** Flow cytometric analysis of CD4 and CD8 expression on thymocytes of WT and KO mice. **d** Percentages of CD4−CD8−(double-negative, DN), CD4+CD8+ (double-positive, DP), CD4+ or CD8+ single-positive (SP) subsets. Mice of 5–8-week old were used. Data represent 10 mice per genotype analyzed in at least three independent experiments (mean ± SEM; unpaired t-test)

the early stages of T-cell differentiation, we crossed mice carrying floxed *Gabpa* alleles (*Gabpa*^f/f)[26] with *CD4*^Cre transgenic mice in which the Cre recombinase is expressed in DP thymocytes. GABPα protein was barely detectable in CD4$^+$ or CD8$^+$ T cells isolated from the spleen and lymph nodes (LNs) of *CD4*^Cre-*Gabpa*^f/f mice, revealing that *Gabpa* was efficiently deleted in both CD4$^+$ and CD8$^+$ T cells in these mice, hereafter designated as knockout (KO) mice (Fig. 1a). Thymic cellularity was comparable between 5- to 8-week-old KO mice and *Gabpa*^f/f littermate controls, hereafter designated as wild-type (WT) mice (Fig. 1b). In addition, the proportions of DN, DP, and CD4 or CD8 single-positive (SP) subsets were comparable between WT and KO mice, with the exception of 20% reduction of CD8SP cells in KO mice (Fig. 1c, d). These observations demonstrate that thymic selection is largely unaltered *in CD4*^Cre*Gabpa*^f/f mice.

**GABPα deficiency impairs peripheral T-cell homeostasis.** To investigate whether GABPα deficiency affects T-cell homeostasis, we enumerated total, CD4$^+$ and CD8$^+$ T cells in peripheral lymphoid organs. Compared to WT mice, GABPα-deficient mice had about twofold reduction of splenic and LN TCRβ$^+$ cells (Fig. 2a–c and Supplementary Fig. 1a–c). Among TCRβ$^+$ cells, CD8$^+$ T cells were more severely affected, as the ratio of CD4$^+$ to CD8$^+$ T cells was higher in KO mice (Supplementary Fig. 1d, e). Further analysis of expression of the T-cell activation marker CD44 revealed that KO mice had reduced fractions of activated or memory phenotype CD4$^+$ or CD8$^+$ T cells (Fig. 2d and Supplementary Fig. 1f). The number of CD44$^{lo}$ naive T cells and CD44$^{hi}$ activated memory T cells was reduced by 2-3-fold and 5–7-fold, respectively, in KO mice (Fig. 2e and Supplementary Fig. 1g). Together, these findings reveal an essential role for GABPα in the maintenance of peripheral T cells.

Naive T-cell homeostasis is dependent on CD62L and IL-7Rα to support T-cell trafficking, survival, and homeostatic proliferation, and their expression is under the control of the forkhead family transcription factor Foxo1[27, 28]. To test whether GABPα is required for naive T-cell survival in vitro, we purified naive CD4$^+$

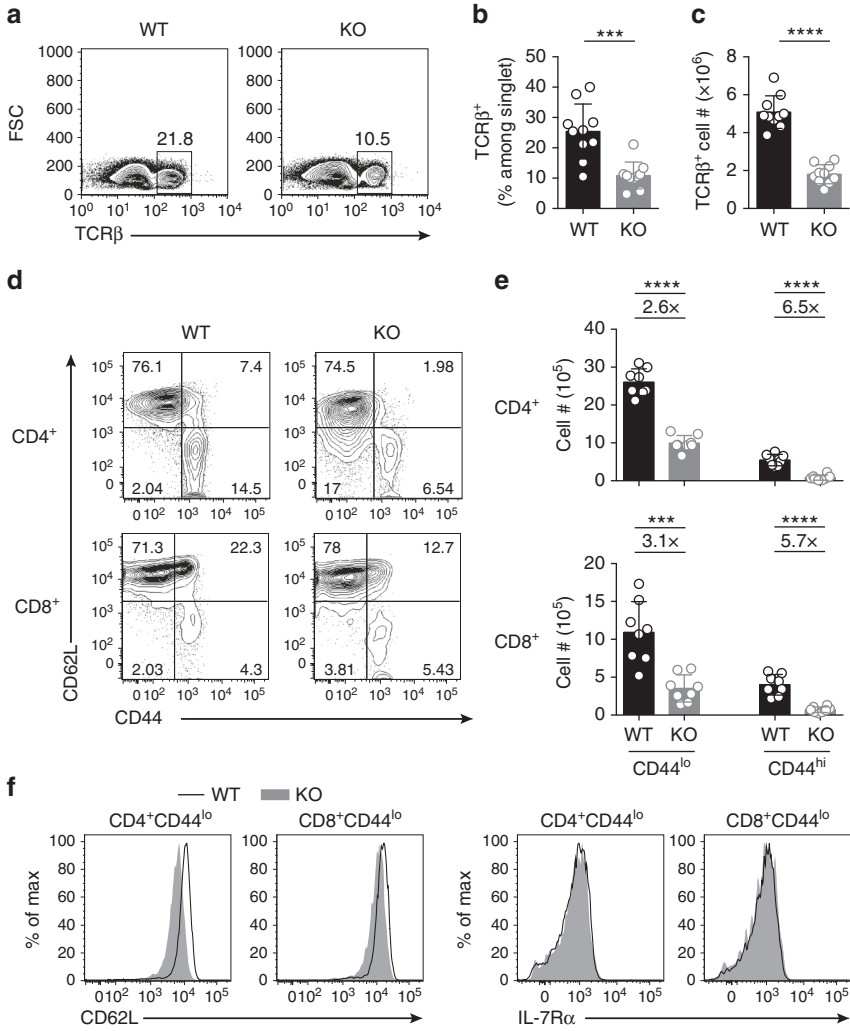

**Fig. 2** Mice with T-cell-specific disruption of GABPα have fewer peripheral T cells. **a** Flow cytometric analysis of TCRβ expression on live singlet cells from spleen of *Gabpa*^f/f (wild-type, WT) and *CD4*^Cre*Gabpa*^f/f (KO) mice. **b, c** Fractions of TCRβ$^+$ cells among total live cells (**b**), and numbers of TCRβ$^+$ cells (**c**) in the spleens of WT and KO mice. **d** Flow cytometric analysis of CD44 and CD62L expression on CD4$^+$ and CD8$^+$ T cells from spleen of WT and KO mice. **e** Numbers of CD44$^{lo}$ and CD44$^{hi}$ subsets of CD4$^+$ and CD8$^+$ in the spleens of WT and KO mice. Fold changes comparing KO to WT are shown above the plots. **f** Expression of CD62L and IL-7Rα in CD44$^{lo}$ naive CD4$^+$ and CD8$^+$ T cells from spleens of WT and KO mice. Mice of 5–8-week old were used. Data represent 8–10 mice per genotype analyzed in at least three independent experiments (mean ± SEM; unpaired *t*-test). ns = not significant. ***$P < 0.001$, ****$P < 0.0001$

and CD8[+] T cells from WT and KO mice, and cultured them with IL-7 containing medium. Although both WT and KO cells retained naive and quiescent phenotype, loss of GABP resulted in compromised cell survival (Supplementary Fig. 2). Previous studies have implicated a role of GABPα in the transactivation of the *Il7r* locus in T cells[18, 25], which might contribute to the GABPα-dependent naive T-cell survival. Surprisingly, while CD44[lo] naive T cells from GABPα-deficient mice expressed modestly lower levels of CD62L than control cells from WT mice (Fig. 2d, f and Supplementary Fig. 1f, h), expression of IL-7Rα was not different between WT and KO T cells (Fig. 2f and Supplementary Fig. 1h). Thus, the regulation of GABPα in T-cell homeostasis is independent of IL-7Rα-mediated mechanisms.

**GABPα deficiency impairs antigen-induced T-cell responses.** In addition to the naive T-cell defects, *CD4^{Cre}Gabpa^{f/f}* mice possessed a more pronounced loss of activated memory T cells (Fig. 2e and Supplementary Fig. 1g). To determine whether GABPα is required for antigen-stimulated T-cell responses, we isolated naive CD4[+] and CD8[+] T cells from WT and KO mice, and stimulated them with CD3 and CD28 antibodies in the presence of IL-2. At 12-h post stimulation, GABPα-replete and -deficient T cells upregulated CD69 and CD25 to a comparable level (Fig. 3a), suggesting that GABPα deficiency did not impair TCR signaling. However, in contrast to WT T cells, KO T cells failed to expand their cell size and further augment the expression of CD69 and CD25 at the 36-h time point (Fig. 3a). In line with the attenuated expression of activation markers, production of IL-2 was abrogated in KO T cells as well (Fig. 3b). Furthermore, GABPα-depleted T cells did not undergo cell division (Fig. 3c), and they were more prone to cell death (Fig. 3d). To further test whether the GABP-mediated regulation is dependent on IL-2 receptor signaling, we stimulated WT and KO T cells in the absence of exogenous IL-2, and similar defects in cell activation, proliferation, and survival in GABP-deficient T cells were observed (Supplementary Fig. 3). Collectively, these observations reveal a crucial function for GABPα in the control of T-cell activation, proliferation, and survival upon antigen challenge.

**GABPα controls a distinct transcriptional program in T cells.** To define the genetic programs by which GABPα controls T-cell responses, we performed transcriptome analysis of GABPα-sufficient and -deficient cells. We purified naive CD4[+] and CD8[+] T cells from WT and KO mice, and either extracted RNA from the sorted T cells (0-h group), or stimulated T cells with anti-CD3 and anti-CD28 in the presence of IL-2 for 18 h before RNA extraction (18-h group). As T-cell homeostasis and antigen-induced responses were affected in both CD4[+] and CD8[+] T cells, we focused on transcripts that had consistent changes between CD4[+] and CD8[+] T cells, and were differentially expressed between WT and KO T cells at 0-h and/or 18-h time points. We uncovered 109 and 80 genes that were downregulated or upregulated, respectively, in KO T cells by more than 1.5-fold and with a false discovery rate <0.05 (Supplementary Data 1).

Gene ontology association analysis revealed that cellular processes including protein and nucleotide metabolism, protein transport and localization, cell cycle progression, cell death regulation, and cellular responses to stress were significantly over-represented in the GABPα-regulated genes (Supplementary Fig. 4 and Supplementary Data 2). Among the genes downregulated in KO T cells, ~70% of them were differentially expressed at both 0- and 18-h time points. These genes encode proteins that are involved in basic cellular activities, such as microtubule organization and vesicle transportation (*Ccz1*, *Nubp1*, and *Mcrs1*), and mitochondrial biogenesis and function (*Stoml2*,

*Romo1*, *Ndufaf1*, *Idh3b*, *Mrpl16*, and *Mrps35*), which could explain the homeostatic defects manifested by both naive and activated T cells lacking GABPα (Supplementary Data 1 and Fig. 2e). Moreover, about 30% of the downregulated genes showed compromised expression in KO T cells only at the 18-h time point (Supplementary Data 1). Many of these genes, including *Tyms*, *Shmt1*, *Pola2*, *Lig1*, *Mcm3*, and *Mcm5*, are important for folate-dependent de novo thymidylate biosynthesis and DNA replication. They were induced in WT cells in response to TCR stimulation, but the induction was abolished in the absence of GABPα (Supplementary Data 1). This additional group of activation-associated genes might be responsible for the exacerbated phenotype exhibited by GABPα-deficient activated T cells compared with the naive counterparts (Fig. 2d, e). On the other hand, the genes that were upregulated in GABP-null cells are mainly involved in ER stress responses and the induction of apoptosis and senescence (*Gadd45g*, *Ddit3*, *Phlda3*, *Pdia5*, *Blcap*, and *Phlda3*), which may explain the survival defects of GABPα-deficient T cells (Supplementary Data 1, Supplementary Fig. 3d, and Supplementary Fig. 2).

We wished to define the direct GABPα-dependent transcriptional program in T cells. To this end, we performed chromatin immunoprecipitation coupled to high-throughput sequencing (ChIP-seq) experiments. Using a cutoff of false discovery rate of 0.01, we uncovered 6782 genomic loci that were enriched in GABPα antibody pulldown compared with the chromatin input (Supplementary Data 3). We found that the GABPα-binding sites were mostly enriched in gene promoter regions (Fig. 4a). De novo motif prediction from the peaks also revealed a conserved GABPα recognition site containing the GGAA core (Fig. 4b). In addition, we could identify previously characterized binding sites of GABPα target genes[29], including the sites in the proximal promoter regions of *Gabpa* and *Cox5b* (Supplementary Fig. 6a). Nonetheless, *Il7r* did not appear to be a GABPα target gene, as WT and KO cells contained comparable amounts of *Il7r* mRNA, and no GABPα-binding peak was found in the *Il7r* gene locus (Supplementary Fig. 5b). These observations are consistent with the ex vivo analysis of IL-7Rα (Fig. 2f and Supplementary Fig. 1h), further supporting IL-7Rα-independent mechanisms of T-cell regulation by GABPα.

By cross-referencing the transcriptome and the ChIP-seq data sets, we could identify 95 putative direct target genes of GABPα, among which 70 and 25 were activated or repressed by GABPα, respectively (Fig. 4c and Supplementary Data 4). Further gene ontology analysis revealed that the GABPα direct target genes were strongly associated with DNA replication, DNA repair, centrosome and cell cycle regulation, RNA processing, protein and sugar metabolism, signal transduction, and transcription (Fig. 4c, Supplementary Fig. 6, and Supplementary Data 5). Among the putative GABPα direct targets are those encoding mitochondrial ribosomal proteins (*Mrpl16* and *Mrps35*), mitochondrial membrane proteins (*Stoml2* and *Tmem126a*), enzymes involved in mitochondrial protein synthesis and quality control (*Mtx2*, *Oma1*, and *Mtif2*), as well as essential components of electron transport chain and oxidative phosphorylation (*Idh3b*, *Ndufaf1*, and *Dlst*) (Fig. 4c). In line with these observations, GABPα-deficient T cells possessed heightened levels of both cellular and mitochondrial reactive oxygen species (ROS) (Supplementary Fig. 7). These findings indicate a critical role for GABPα in the control of cellular metabolism and redox balance, the deficiency of which may affect T-cell homeostasis.

**GABPα promotes *Mcm* expression and DNA replication in T cells.** Considering that several GABPα target genes were involved in cell cycle regulation and GABPα-deficient T cells

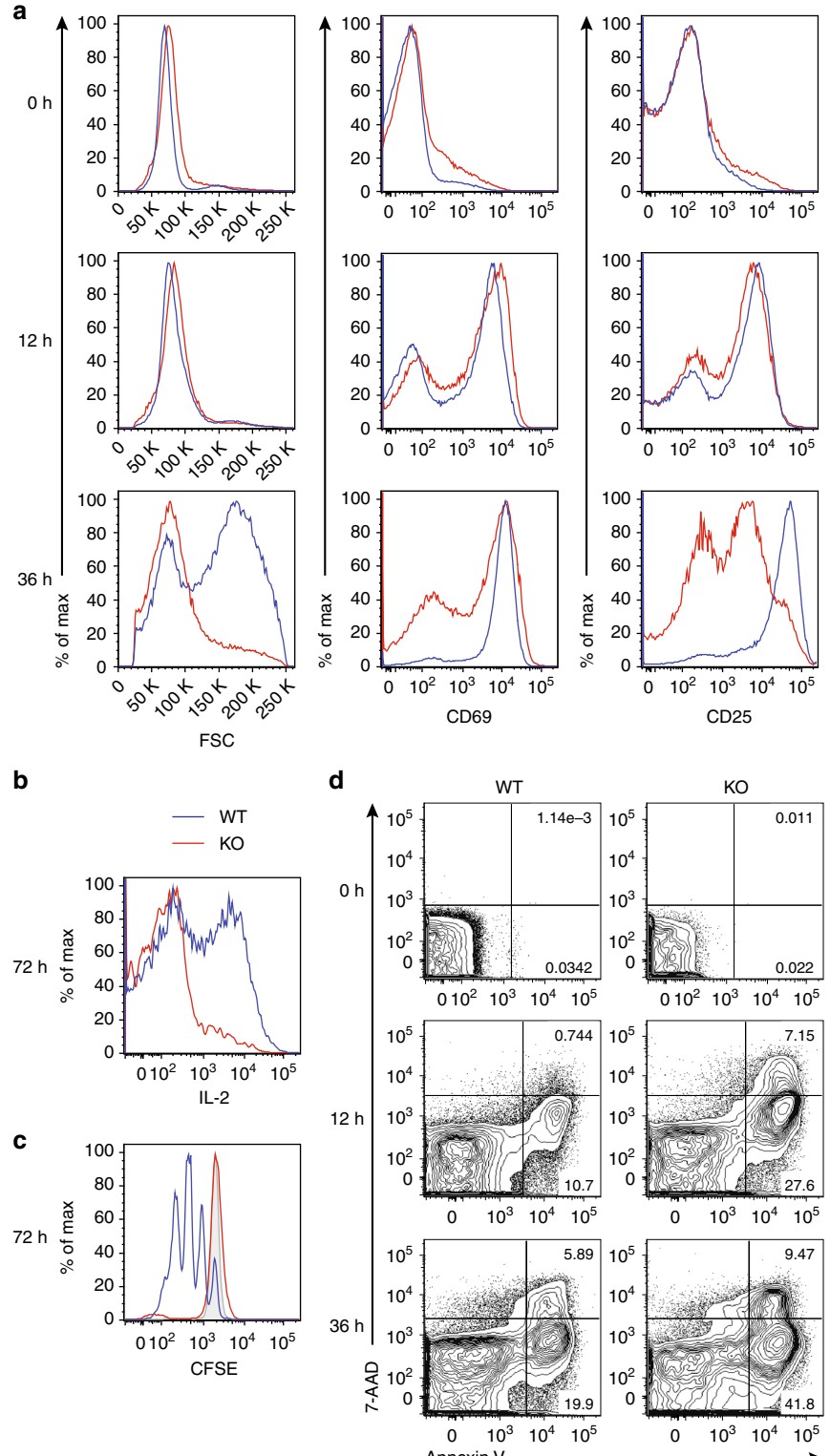

**Fig. 3** GABPα is required for T-cell responses to antigen stimulation in vitro. Naive (CD62L[hi] CD44[lo]) CD4[+] or CD8[+] T cells from *Gabpa*[f/f] (WT) and *CD4*[Cre]*Gabpa*[f/f] (KO) mice were purified by flow cytometric sorting, and were subjected to anti-CD3 and anti-CD28 stimulation in the presence of IL-2. Representative plots from CD8[+] T-cell culture were shown. **a** Analysis of cell size (FSC) and activation markers, CD69 and CD25, at 0, 12, and 36 h post stimulation. **b** Seventy-two hour after cell culture, WT and KO cells were restimulated with PMA and ionomycin for 4 h and analyzed for the expression of IL-2 by intracellular cytokine staining. **c** WT and KO cells were labeled with the cytosolic dye CFSE, and cell division was assessed by the dilution of CFSE. Gray shaded line shows CFSE level of undivided cells. **d** Cell death was assessed with Annexin V and 7-AAD staining. Data represent at least three independent experiments

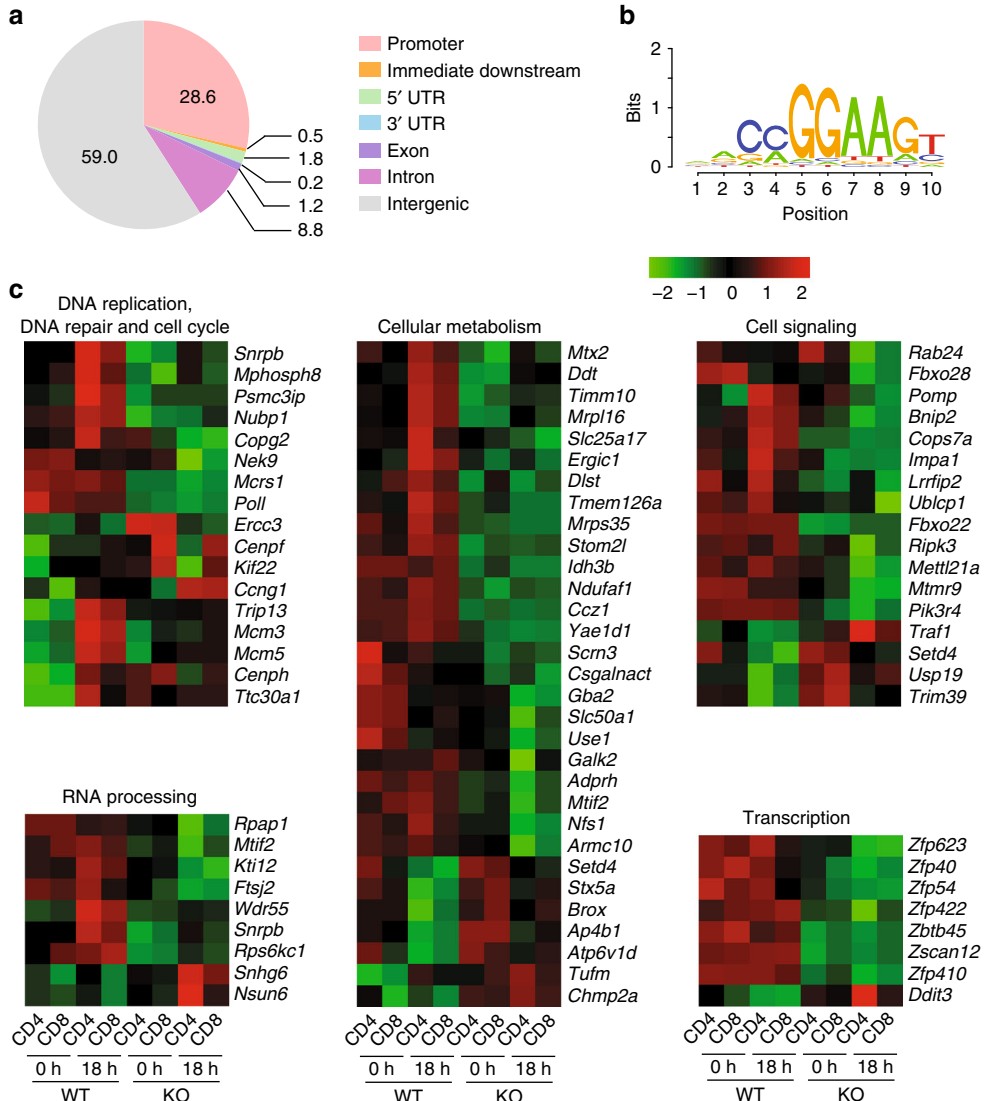

**Fig. 4** GABPα-dependent transcriptional program in T cells. **a** Pie chart of distribution of GABPα-binding peaks discovered from ChIP sequencing. **b** The consensus sequence motif identified in the GABPα-binding sites by the HOMER program. **c** Heat map of GABPα direct target genes in WT and GABPα-deficient CD4+ and CD8+ T cells at two different time points: 0-h, sorted (CD62LhiCD44lo) naive T cells; 18-h, sorted naive T cells were subjected to anti-CD3 and anti-CD28 stimulation in the presence of IL-2 for 18 h. GABPα direct target genes were defined as: (1) differentially expressed between WT and GABPα knockout T cells at 0- or 18-h time point or both; (2) expression changes were consistent between CD4+ and CD8+ T cells; (3) GABPα was recruited to the gene locus in the ChIP-seq experiment. The genes were divided into five groups on the basis of gene ontology

failed to proliferate (Figs. 4c and 3c), we wished to determine the exact stage of GABPα-dependent cell cycle progression in T cells. To this end, we stimulated WT and KO T cells with CD3 and CD28 antibodies, and pulse-labeled T cells with EdU, a nucleoside analog of thymidine, which can be incorporated into the replicating DNA. Strikingly, while ~20% WT cells incorporated EdU, less than 1% GABPα-deficient T cells did so (Fig. 5a). These observations demonstrate that GABPα promotes S phase transition in T cells.

Among the GABPα direct target genes, *Mcm3* and *Mcm5* have been implicated in the regulation of S phase entry (Fig. 4c). The minichromosome maintenance (Mcm) proteins are a family of highly conserved proteins that form a hexameric complex (Mcm2-7), which functions as a replicative helicase crucial for the initiation and elongation of DNA replication in eukaryotes[30]. TCR and CD28 stimulation of WT T cells induced the expression of Mcm3 and Mcm5, but this induction was diminished in

GABP-deficient T cells (Fig. 4c). In addition, GABPα was recruited to the promoters of *Mcm3* and *Mcm5* (Fig. 5b), with evolutionarily conserved GABP-binding elements identified within these regions (Fig. 5c). Notably, the proximal promoter region of *Mcm3* contained two GABP motifs that were 10-bp apart (Fig. 5c), which aligned well with the 10.5-bp periodicity of double-helical B-DNA and might act synergistically to recruit a GABPα2β2 tetrameric complex[14]. Indeed, ChIP-qPCR experiments validated the binding of GABPα to *Mcm3* and *Mcm5* promoters in both CD4+ and CD8+ T cells (Fig. 5d). We further tested the functional relevance of Mcm3 in the GABPα-dependent cell cycle regulation via small hairpin RNA (shRNA) knockdown experiments (Supplementary Fig. 8). Knocking down Mcm3 in WT T cells resulted in reduced incorporation of EdU in response to anti-CD3 and anti-CD28 stimulation, which partially phenocopied the loss of GABPα (Supplementary Fig. 8).

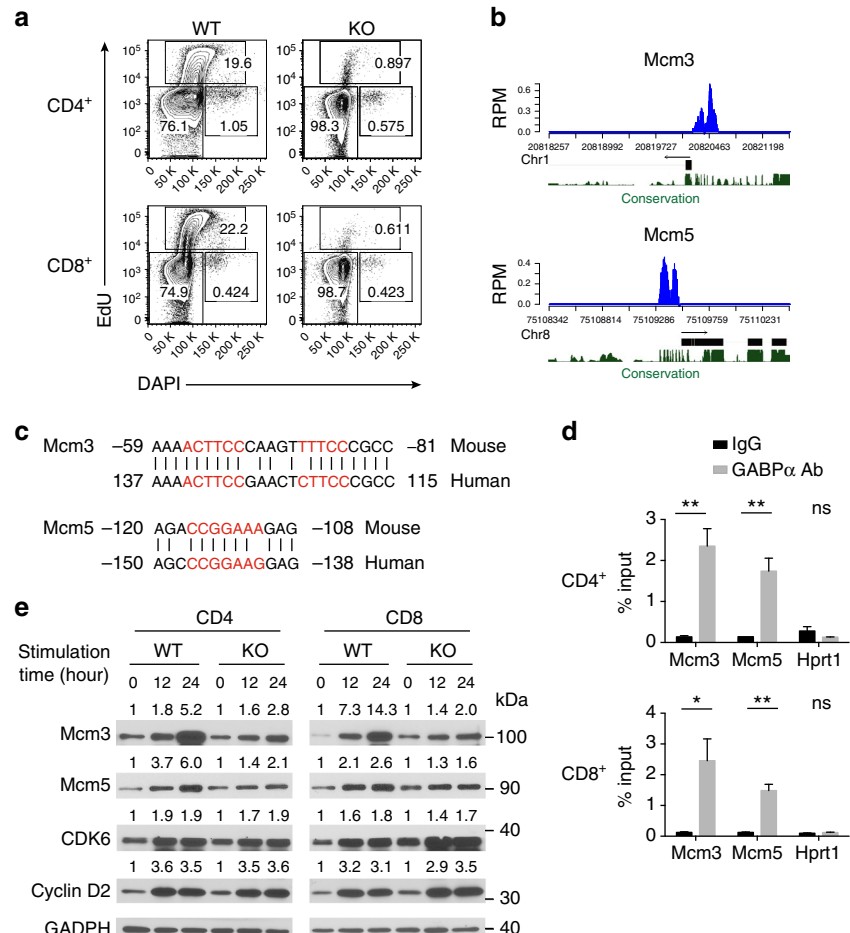

**Fig. 5** GABPα regulates DNA replication and cell cycle progression. **a** Naive (CD62L$^{hi}$ CD44$^{lo}$) CD4$^+$ or CD8$^+$ T cells from *Gabpa$^{f/f}$* (WT) and *CD4$^{Cre}$Gabpa$^{f/f}$* (KO) mice were purified by flow cytometric sorting, and were subjected to anti-CD3 and anti-CD28 stimulation in the presence of IL-2. After 22 h, cells were pulsed with EdU for 2 h. Flow cytometric analysis of DAPI and EdU was shown. **b** GABPα-bound regions in the *Mcm3* and *Mcm5* gene loci. Gene structure, chromosomal location, and sequence homology were shown below the binding peaks. The *Y*-axis represent read per million (RPM). **c** Alignment of the conserved GABPα-binding sites in mouse and human *Mcm3* and *Mcm5* proximal promoter regions. The consensus GABPα-binding sequences were marked in red, and nucleotides were numbered relative to the transcription start site. **d** ChIP-qPCR validation of GABPα binding to the promoter regions of *Mcm3* and *Mcm5* genes in WT CD4$^+$ T cells and CD8$^+$ T cells. IgG was used as a control for the GABPα antibody. *Hprt1* locus was used as a negative control. Bars are mean ± SEM with unpaired *t*-test, ns = not significant, *$P < 0.05$, **$P < 0.01$. **e** Naive CD4$^+$ or CD8$^+$ T cells from WT and KO mice were purified by flow cytometric sorting, and were stimulated with CD3 and CD28 antibodies in the presence of IL-2 for 0, 12, or 24 h. Immunoblotting analysis of Mcm3, Mcm5, CDK6, and Cyclin D2 was shown. GADPH was used as a loading control. The relative amounts of proteins were normalized to the 0-h time point of each individual group. Uncropped western blotting images were included in Supplementary Fig. 10

In addition to the Mcm complex, another major component of cell cycle regulation is the ordered expression of cyclins and cyclin-dependent kinases (CDKs)[31]. D-type cyclins and their catalytic partners, CDK4 and CDK6, are the first integrators of extracellular stimuli[32]. Following their induction in early G1 phase, cyclin D-CDK4/6 complexes phosphorylate key substrates, including Rb, leading to the release of E2F transcription factors and transition into S phase[32]. Intriguingly, while enhanced expression of Mcm3 and Mcm5 proteins was observed in WT cells following T-cell activation, expression of these proteins was greatly attenuated in GABPα-null T cells (Fig. 5e). In contrast, the induction of cyclin D2 and CDK6 was not affected by the loss of GABPα (Fig. 5e), demonstrating that GABPα controls the cell cycle entry of T cells by specifically modulating the Mcm-dependent pathway.

**Cell-intrinsic role of GABPα in T-cell homeostatic proliferation.** Having established a critical role of GABPα in T-cell cycle

progression in vitro, we wished to investigate whether GABPα was required for T-cell homeostatic proliferation in vivo. To this end, we purified naive CD4$^+$ and CD8$^+$ T cells from congenically marked WT and KO mice, mixed them at 1:1 ratio, and labeled them by CFSE staining. We transferred these cells intravenously into Rag1-deficient or the sublethally irradiated C57/BL6 recipients (Fig. 6a), and assessed cell proliferation 7 days post transfer. In both Rag1-deficient and the irradiated C57/BL6 recipients, KO T cells exhibited reduced proliferation, and were out-competed by their WT counterparts (Fig. 6b). These findings indicate that GABPα deficiency impairs T-cell homeostatic proliferation in vivo.

**GABPα is required for antigen-induced T-cell responses in vivo.** To determine the function of GABPα in antigen-induced T-cell responses in vivo, we used a *Listeria monocytogenes* infection model. We used the *L. monocytogenes* strain that expresses ovalbumin (OVA) as a model antigen (LM-OVA).

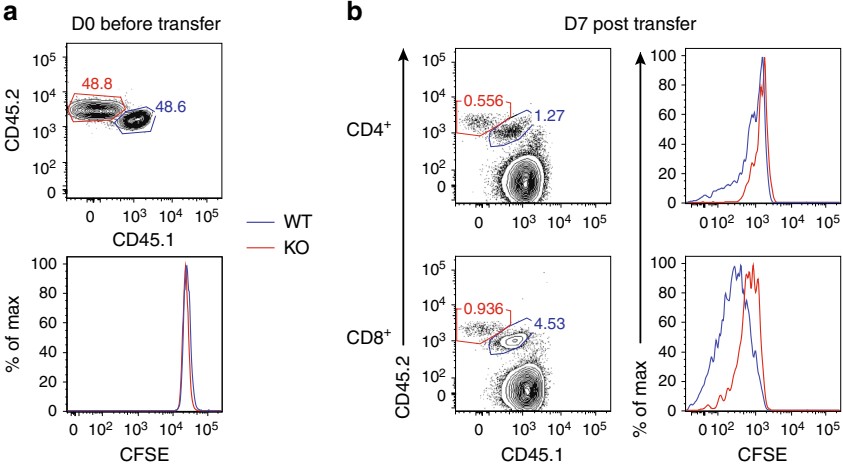

**Fig. 6** A cell-intrinsic role of GABPα in control of T-cell homeostatic proliferation in vivo. CD4[+] or CD8[+] T cells from WT (CD45.1/CD45.2) and KO (CD45.2/CD45.2) mice were mixed at a 1:1 ratio and transferred into Rag[−/−] or sublethally irradiated C57/B6 recipients (CD45.1/CD45.1). **a** The WT and KO CD8[+] T-cell populations before transfer. The staining of the cytosolic dye CFSE was comparable between WT and KO cells. **b** Representative flow cytometric plots of transferred WT and KO cells in sublethally irradiated C57/B6 recipients at day 7 post transfer. Data represent at least three independent experiments

Clearance of the bacteria is mediated by T cells, among which CD8[+] T cells provide the most substantial contribution to the protective immunity[33]. We crossed *Gabpa*[f/f] mice to the OT-1 transgenic background to generate OVA-specific CD8[+] T cells (OT-1 cells), and subsequently bred the *Gabpa*[f/f]*OT-1* mice with mice harboring the *CD8*[Cre] transgene. GABPα-replete and -depleted OT-1 cells were purified from cogenically marked mice, mixed at 1:1 ratio and stained with CFSE dye (Fig. 7a). T cells were transferred to recipient mice expressing a different congenic marker, which were subsequently challenged with a sublethal dose of LM-OVA. OT-1 cell responses in the spleen and liver, two major target organs of *L. monocytogenes* infection, were analyzed at days 3 and 7 post transfer. Compared with the WT OT-1 cells, a substantially lower proportion of KO OT-1 cells underwent cell division at day 3 (Fig. 7b, d). As a consequence, KO OT-1 cells were markedly out-numbered by WT OT-1 cells at day 7 (Fig. 7c, e).

Given the striking defects GABPα-null OT-1 cells exhibited upon LM-OVA infection, we next examined the endogenous T-cell responses in *CD4*[Cre]*Gabpa*[f/f] mice. Seven days after the LM-OVA inoculation, a sizeable OVA-specific (K[b]-ova[+]) CD8[+] T-cell population was found in WT mice (Fig. 7f, g). However, this antigen-specific subset was barely detectable in *CD4*[Cre]*Gabpa*[f/f] mice (Fig. 7f, g). The very limited pool of OVA-specific CD8[+] T cells in *CD4*[Cre]*Gabpa*[f/f] mice was composed of comparable percentages of IL-7Rα[hi]KLRG1[lo] memory-precursor effector T cells and IL-7Rα[lo]KLRG1[hi] short-lived effector T cells as their WT counterparts (Supplementary Fig. 9a, b). In addition, compared with GABPα-replete cells, GABPα-deficient K[b]-ova[+] cells expressed similar levels of the cytotoxic molecule, granzyme B, as well as comparable or slightly reduced levels of the pro-inflammatory cytokine IFN-γ (Supplementary Fig. 9c, d). Taken together, these observations demonstrate that GABPα plays a crucial, cell-autonomous role in promoting antigen-specific T-cell responses.

## Discussion
GABP has been shown to regulate early T-cell development[18, 25], yet its function in mature T cells has not been studied. Using a *CD4*[Cre] conditional knockout system, we found that GABPα deficiency substantially impaired peripheral T-cell homeostasis

while thymic development was largely unperturbed. Notably, the reduction of naive T cells was independent of IL-7Rα, which is a previously documented target of GABPα[18]. In addition, GABPα-deficient T cells showed compromised activation, proliferation, and survival in response to antigenic stimulation in vitro. Mechanistically, by coupling transcriptome analysis with ChIP-seq, we identified GABPα as a vital regulator of cellular redox homeostasis, DNA replication, and cell cycle progression. Subsequent experiments further established the importance of GABPα-dependent transcriptional program in the control of T-cell-mediated adaptive immunity in vivo, as depletion of GABPα undermined T-cell homeostatic proliferation in lymphopenic environment as well as antigen-induced responses to microbial infection.

A surprising yet critical finding of this present study is that GABPα is superfluous for IL-7Rα expression in mature T cells. Based on an earlier study, GABPα bound to a GGAA motif in the *Il7r* 5′ regulatory region, and knockdown of GABPα reduced IL-7Rα expression in vitro[18]. However, we found that neither the mRNA or protein level of *Il7r* was perturbed by GABPα ablation in mature T cells. Nor did we observe any GABPα-binding peak in the *Il7r* locus from our ChIP-seq. One plausible explanation for this discrepancy is the redundancy among the large Ets family transcription factors. It is possible that other Ets proteins that are highly expressed in peripheral T cells compete with GABPα in DNA binding, since they recognize and bind similar cognate DNA sequences[6]. Indeed, several genome-wide ChIP experiments revealed co-occupancy of the same genomic regions by multiple Ets proteins[34–36]. An alternative explanation is that other transcription factors and cofactors are recruited to the *Il7r* regulatory region in peripheral T cells, thus precluding the access of GABPα. Consistent with this notion, Foxo1 has been shown to bind a conserved site in the *Il7r* proximal promoter[28], which overlaps with the proposed GABPα-binding motif[18]. Moreover, the transcriptional repressor Gfi-1 has also been implicated in the regulation of *Il7r* expression[37]. The functional interaction between GABPα and these transcription factors awaits further investigation.

Our findings here demonstrate that GABPα facilitates homeostatic proliferation of naive T cells as well as clonal expansion of antigen-specific T cells. On the molecular level, GABPα regulates multiple genes involved in DNA replication and

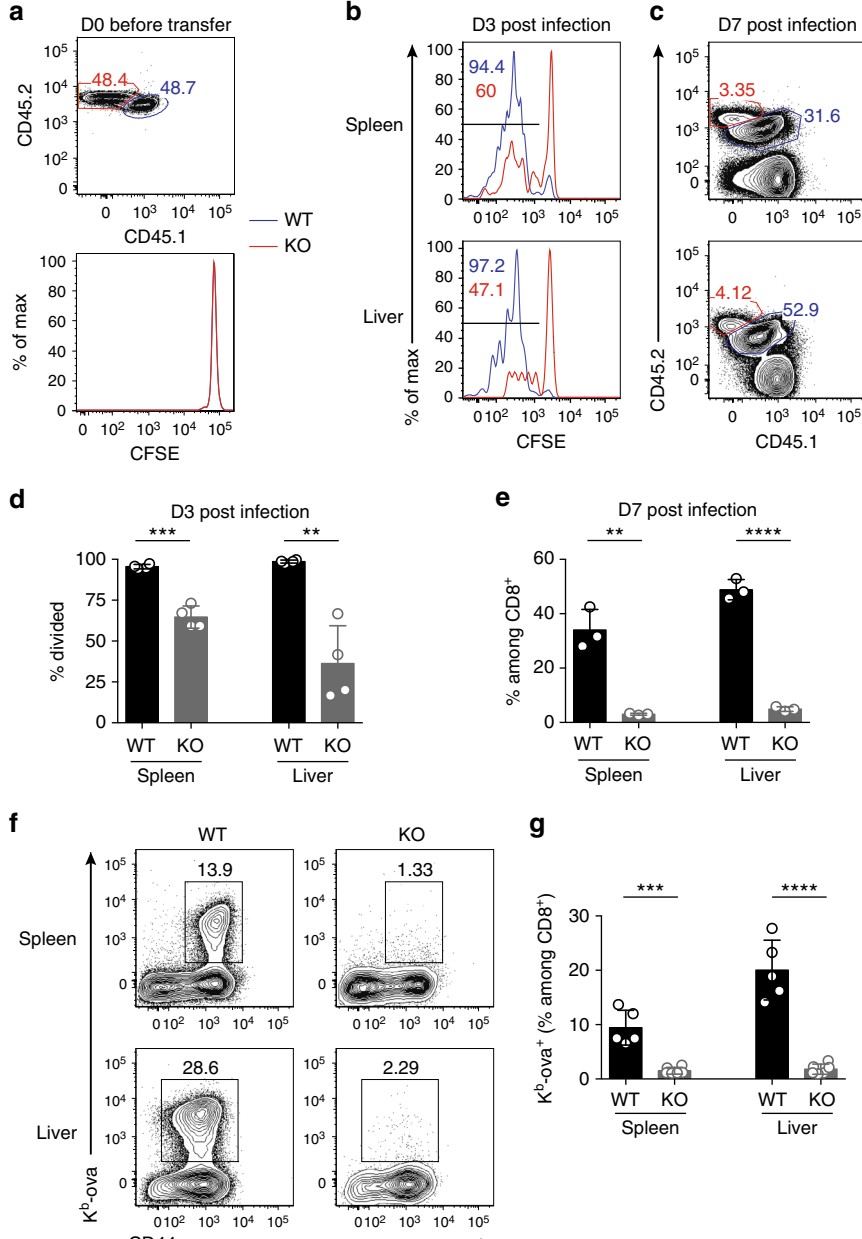

**Fig. 7** GABPα controls T cell responses to *L. monocytogenes* infection. **a–e** OT-1 cells from *Gabpa^{f/f}OT-1* (WT, CD45.1/CD45.2) and *CD8^{Cre}Gabpa^{f/f}OT-1* (KO, CD45.2/CD45.2) mice were mixed at a 1:1 ratio and transferred into WT recipients (CD45.1/CD45.1). The recipients were infected with LM-OVA a day after the cell transfer. **a** The WT and KO OT-1 T cells before transfer. **b** Dilution of CFSE at day 3 post infection. **c** Representative flow cytometric plots of transferred WT and KO cells at day 7 after inoculation. **d** Fractions of cells that have diluted CFSE at D3 p.i. **e** Enumeration of WT and KO cells at D7 p.i. 4 mice (D3 p.i.) or 3 mice (D7 p.i.) were included. **f, g** *Gabpa^{f/f}* (WT) and *CD4^{Cre}Gabpa^{f/f}* (KO) mice were infected with LM-OVA. At day 7 post infection, OVA-specific CD8+ T-cell responses were analyzed by the staining of K^b-ova tetramer. **f** Representative flow cytometric blots of CD44 and K^b-ova staining in CD8+ T cells from spleen and liver. **g** Fractions of K^b-ova+ T cells among the CD8+ population. Data represent three–five mice per genotype analyzed in at least three independent experiments. Bars are mean ± SEM with an unpaired *t*-test, ns = not significant, **$P < 0.01$, ***$P < 0.001$, ****$P < 0.0001$.

cell cycle progression, among which include the *Mcm* genes. Proper assembly, loading, and activation of the Mcm2-7 helicase complex are crucial for DNA replication[30]. Because yeast contains stable levels of Mcm mRNAs and proteins during cell cycle, the majority of studies have focused on the loading and subsequent activation of Mcm complexes rather than transcriptional regulation of *Mcm* genes[30, 38]. Nevertheless, several studies revealed that mammalian cells show a dramatic upregulation of Mcm mRNAs in response to mitogenic cues, such as serum stimulation of resting fibroblasts and stimulation of human primary T cells with

CD3/CD28 antibodies[39, 40]. Silencing *Mcm* genes in T cells causes premature chromatid separation and gross genomic instability[39], implying that although poorly understood, the transcriptional mechanism of Mcm expression is, at least in part, central to the regulation of Mcm activities. To date, E2F has been the only transcription factor implicated in the *Mcm* transcription[41, 42]. In this report, we identified GABPα as another key transcription activator of the *Mcm* genes in T cells. Our future work will explore whether GABPα collaborates with the E2F transcription factors in this regulation and how it occurs.

In fact, GABPα-mediated cell cycle control has been implicated in previous studies. A report on mouse embryonic fibroblasts showed that GABP modulates the expression of *Tyms*, *Skp2*, and *Pola1* through a cyclin D-CDK4/6-independent pathway[16]. In line with this report, we found that GABPα-deficient T cells contained reduced levels of *Tyms*, *Skp2*, and *Pola1* mRNAs, while the expression of cyclin D2 or CDK6 was not affected. Additionally, we identified weak binding peaks of GABPα in the promoter regions of *Tyms*, *Skp2*, and *Pola1*, although their intensity was lower than our stringent cutoff. These findings indicate that GABPα-mediated cell cycle regulation might be broadly applicable. Indeed, cell cycle blockade caused by GABPα depletion was observed in other cell types, including hematopoietic stem cells, vascular smooth muscle cells, and a human liver carcinoma cell line[16, 22, 23, 43].

Besides the cell cycle arrest, GABPα-deficient T cells are more prone to cell death. The survival defects can be partially attributed to the abortive cell division, yet additional GABPα-mediated programs can contribute to this phenotype. We found that various cellular metabolic processes were deregulated in GABPα-deficient T cells, which was associated with disrupted antioxidant defense and reduced cellular fitness. In line with our observations, knockdown or depletion of GABPα triggered augmented cell death in mouse hematopoietic stem cells and a human liver carcinoma cell line, despite that different mechanisms were proposed[23, 29]. Initially identified as nuclear respiratory factor 2, GABP has been shown to control both mitochondrial DNA genes and nuclear genes[10, 15, 44]. Although we did not observe a loss of mitochondria mass (data not shown), numerous components of the tricarboxylic acid cycle and electron transport chain were expressed at low levels in GABPα-deficient T cells. We will elucidate the impact of GABPα on different metabolic programs in mature T lymphocytes in our future studies.

In conclusion, in this report we have identified a critical role for GABP in naive T-cell homeostasis and antigen-stimulated effector responses. This was mediated in part by GABP control of cell cycle progression and cellular metabolism. Manipulation of the GABP pathway may provide novel therapeutic strategies for T-cell-mediated disorders, including infectious diseases and autoimmunity.

## Methods

**Mice**. All mice were on C57BL/6 background. The *GABPa^{f/f}* mouse strain was kindly provided by Dr Steve Burden (New York University)[26]. The *CD4-Cre* and OT-1 transgenic mice were described previously[28, 45]. *CD8^-Cre*[46] and CD45.1^+ mice were purchased from Jackson Laboratory. In all experiments, littermate controls were used when possible. Both male and female mice were included. All mice were maintained under specific pathogen-free conditions, and all animal experimentation was approved by the Institutional Animal Care and Use Committee of Memorial Sloan Kettering Cancer Center.

**Cell isolation**. After whole-body perfusion with 50 ml of heparinized PBS, lymphocytes were isolated as follows. Single-cell suspensions were prepared from spleens and peripheral (axillary, brachial, and inguinal) lymph nodes by tissue disruption with glass slides. To isolate cells from the liver, tissues were finely minced and digested with 1 mg/ml Collagenase D (Worthington) for 30 min at 37 °C. After the digestion, cells were filtered through 70 μM cell strainer, layered in a 44 and 66% Percoll gradient (Sigma), and centrifuged at 3000 rpm for 30 min without brake. Cells at the interface were collected and analyzed by flow cytometry.

**T-cell culture**. Naive (CD25^-CD62L^{hi}CD44^{lo}) CD4^+ and CD8^+ T cells were purified from spleen and lymph nodes of *Gabpa^{f/f}* (WT) and *CD4^{Cre}Gabpa^{f/f}* (KO) mice by flow cytometry sorting (BD FACS Aria). Sorted T cells were cultured with plate-bound α-CD3 (Clone 145-2C11, coated overnight, 5 μg/ml), soluble α-CD28 (Clone: 37.51, 2 μg/ml) and IL-2 (100 U/ml), or with IL-7 (10 ng/ml) for indicated time periods. For EdU incorporation experiments, the cells stimulated with were α-CD3/28 and IL-2 for 22 h and EdU (10 μM) was added into the culture for 2 h.

**shRNA knockdown**. Short hairpin RNA targeting Mcm3 or scramble sequences were packed into a retroviral vector expressing GFP, and transfected into

HEK293T cells with calcium phosphate. Naive CD8^+ T cells isolated from WT C57BL/6 mice were activated with plate-bound α-CD3 (coated overnight, 5 μg/ml), soluble α-CD28 (2 μg/ml), and IL-2 (100 U/ml) for 18 h and were transfected with shRNA expressing retrovirus and sorted based on their GFP expression. Cells were rest for 18 h prior to EdU incorporation assessment. The shRNA sequence for Mcm3 is: GATTGCCTGTAATGTGAAGCAGATGAGTA.

**Adoptive transfer of T cells**. CD4^+ or CD8^+ T cells from *Gabpa^{f/f}* (WT, CD45.1/CD45.2) and *CD4^{Cre}Gabpa^{f/f}* (KO, CD45.2/CD45.2) mice were purified by flow cytometric sorting (BD Aria 2), mixed at a 1:1 ratio, stained with CFSE (5 μM final concentration at room temperature for 5 min), and transferred into Rag-deficient or sublethally irradiated WT recipients (CD45.1/CD45.1) via intravenous injection. A total number of 2 × 10^6 cells were transferred into each recipient. Spleen and lymph nodes were analyzed 7 days after transfer.

For OT-1 transfer experiment, OT-1 cells from *Gabpa^{f/f}OT-1* (WT, CD45.1/CD45.2) and *CD8^{Cre}Gabpa^{f/f}OT-1* (KO, CD45.2/CD45.2) mice were purified by flow cytometric sorting (BD Aria 2), mixed at a 1:1 ratio, stained with CFSE (5 μM final concentration), and intravenously transferred into WT recipients (CD45.1/CD45.1). A total number of 1.5 × 10^5 OT-1 cells were transferred (7.5 × 10^4/genotype).

**Listeria monocytogenes infection**. For the study of primary immune response, mice were intravenously infected with 5 × 10^3 colony-forming units (cfu) of *Listeria monocytogenes* expressing ovalbumin (LM-OVA), and spleens and livers were isolated for analysis 7 days after infection. For OT-1 transfer experiment, mice that had received OT-1 cells were intravenously infected with 1 × 10^5 cfu of LM-OVA 1 day after the adoptive T-cell transfer. Spleens and livers of the infected mice were analyzed 3 and 7 days post infection.

**Flow cytometry**. Fluorochrome-conjugated antibodies against CD45.1 (clone 104), CD45.2 (A20), TCR-β (H57-595), CD4 (RM4-5), CD8 (17A2), CD25 (PC61.5), CD44 (IM7), CD62L (MEL-14), CD69 (H1.2F3), IL-2 (JES6O5H4), IL-7Rα (A7R34), IFN-γ (XMG1.2), and KLRG-1 (2F1) were purchased from eBioscience. Anti-GzmB (GZ11) was purchased from Invitrogen. All antibodies were tested with their respective isotype controls. PE-conjugated K^b-ova tetramer was obtained from the Tetramer Core Facility at Memorial Sloan Kettering Cancer Center. Cell surface staining was performed by incubating cells with specific antibodies for 30 min on ice in the presence of 2.4G2 mAb to block FcγR binding. IL-2 staining was carried out using the intracellular cytokine staining kits from BD Biosciences. To determine cytokine expression, isolated cells were stimulated with 50 ng/ml phorbol 12-myristate 13-acetate (Sigma), 1 mM ionomycin (Sigma), and GolgiStop (BD Biosciences) for 4 h prior to staining. For IFN-γ expression of LM-OVA-infected mice, lymphocytes were stimulated with 10 nM SIINFEKL peptide in the presence of GolgiStop (BD) for 5 h at 37 °C. Apoptotic cell death staining was performed with Annexin V staining kit (BD) according to the manufacturer's instructions. Cellular ROS and mitochondrial ROS levels were determined by 2',7'-dichlorodihydrofluorescein diacetate (H2DCFDA) and MitoSOX red staining, respectively (ThermoFisher). Incorporation of EdU was measured using the Click-iT EdU flow cytometry assay kit (Invitrogen). For all stains, dead cells were excluded from analysis by means of Live/Dead Fixable Dye (Invitrogen), DAPI, or propidium iodide (PI) stain. All samples were acquired and analyzed with LSRII flow cytometer (Becton Dickson) and FlowJo software (TreeStar).

**Gene-expression profiling**. Naive (CD25^-CD62L^{hi}CD44^{lo}) CD4^+ and CD8^+ T cells were purified from spleen and lymph nodes of *Gabpa^{f/f}* (WT) and *CD4^{Cre}Gabpa^{f/f}* (KO) mice by FACS sorting. Half of the sorted naive T cells was used for the 0-h time point, whereas the other half of cells was subject to α-CD3/28 and IL-2 stimulation for 18 h. At indicated time points, cells were lysed using QIAZol reagent (Qiagen), followed by RNA extraction with the miRNeasy Mini Kit according to the manufacturer's instructions (Qiagen). Two rounds of RNA amplification, labeling, and hybridization to M430 2.0 chips (Affymetrix) were carried out at the Genomics Core of Memorial Sloan Kettering Cancer Center. Gene-expression profiling analyses were done with R statistical environment. Affymetrix CEL files of microarray experiments were processed using the "affy" package and differential expression was assessed using "limma"[47] package of the Bioconductor Suite (http://www.bioconductor.org/). A linear model was fitted to each gene, and empirical Bayes moderated *t*-statistics were used to assess differences in expression using limma package[47]. Empirical Bayes moderated-*t* P values were computed relative to a fold-change cutoff of 1.5-fold using WT samples. Genes were considered differentially expressed (GABPα-dependent) if they had an FDR less than 0.05.

**ChIP-seq**. CD4^+ T cells of WT mice were purified by MACS beads (Miltenyi) and fixed for 10 min at room temperature with 10% formaldehyde. Glycine was added to a final concentration of 0.125 M to quench the formaldehyde. Cells were pelleted, washed twice, and ice-cold PBS and lysed with hypotonic lysis buffer. Chromatin was sheared with Bioruptor sonicator (Diagenode) to 200–500 base pairs (bp) in length. The prepared chromatin was incubated with 5 μg anti-GABPα (sc-22810X, Santa Cruz). Immune complexes were washed and eluted. Precipitated

DNA ChIP DNA and input DNA were incubated at 65 °C to reverse the cross-linking. After digestion with RNase and proteinase K, the ChIP and input DNA were purified with phenol/chloroform extraction and ethanol precipitation. The purified DNA was repaired, ligated with adapter, and amplified by PCR for 15–20 cycles. The amplified DNA was size selected by gel extraction and used for sequencing. Sequencing (36-bp single-end) was performed at the Genomics Core of Memorial Sloan Kettering Cancer Center using HiSeq (Illumina). Reads were first processed with trimmomatic to remove the adapter sequences and bases with quality scores below 20, and reads with less than 30 remaining bases were discarded[48]. Trimmed reads were then aligned to mm10 mouse genome with the bowtie aligner[49]. GABP peaks were called using MACS2 using $q$ value cutoff 0.01[50]. The distribution of the peaks around the TSS was calculated using the ChIPpeakAnno package[51]. DNA motif analysis was performed with the Hypergeometric Optimization of Motif Enrichment program (HOMER)[52].

**ChIP-qPCR.** CD4[+] T cells and of CD8[+] T cells from WT mice were purified and processed as described in ChIP-seq. The prepared chromatin was incubated with 5 μg anti-GABPα (sc-22810X, Santa Cruz) or control rabbit immunoglobulin (2729, Cell Signaling) overnight. Precipitated DNA from the GABPα antibody and IgG control groups was analyzed by PCR with the following primers:
  *Mcm3*: 5′-CAGAGAGGACGCTCAAAACC-3′ and 5′-AGAAAAACAGGGGG TGAGGT-3′;
  *Mcm5*: 5′-CAAAATGGAGACCGGAAAGA-3′ and 5′-GCGGATAGCTATTG GACTGC-3′;
  *Hprt1*: 5′-TGAGCGCAAGTTGAATCTG-3′ and 5′-GGACGCAGCAACTGAC ATT-3′.

**Gene ontology analysis.** The BiNGO 3.0.3 plugin for cytoscape was used to determine which gene ontology categories were statistically enriched for the GABPα-regulated genes or GABPα direct target genes identified from the microarray and ChIP-seq experiments[53].

**Immunoblotting.** CD4[+] and CD8[+] T cells were purified from spleen and lymph nodes of *CD4^Cre Gabpa^f/f* mice and littermate controls by FACS sorting (BD, Aria). In the time course culture experiment, naive T cells were purified, subject to α-CD3/28 and IL-2 culture, and collected at indicated time points. Total protein extracts were dissolved in SDS sample buffer, separated on 12% SDS-PAGE gels and transferred to polyvinylidene difluoride membrane (Millipore). The membranes were probed with antibodies against GABPα (sc-22810, Santa Cruz), CDK6 (3136, Cell Signaling), Cyclin D2 (3741, Cell Signaling), Mcm3 (4003, Cell Signaling), Mcm5 (A300-195A, Bethyl Laboratories), GADPH (ab9485, Abcam), and β-actin (AC-15, Sigma), and visualized with the Immobilon Western Chemiluminescent HRP Substrate (Millipore).

**Statistical analysis.** All data are presented as the mean values ± SEM. Comparisons between groups were analyzed using unpaired Student's *t*-tests. ns = not significant, *$P < 0.05$, **$P < 0.01$, ***$P < 0.001$, ****$P < 0.0001$.

**Data availability.** Microarray and ChIP-seq data that support the findings of this study have been deposited in GEO with the primary accession code GSE101937.

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

## Acknowledgements

We thank members of the Li lab for helpful discussions and critical reading of the manuscript. This work was supported by the National Institutes of Health (RO1 AI102888-01A1 to M.O.L.), a Faculty Scholar Award from the Howard Hughes Medical Institute (M.O.L.), and the Memorial Sloan Kettering Cancer Center Support Grant/Core Grant (P30 CA008748).

## Author contributions

C.T.L. and M.O.L. were involved in all aspects of this study including planning and performing experiments, analyzing and interpreting data, and writing the manuscript. M.H.D., M.R.B., A.T., D.K. and Y.X. assisted with experiments. H.U.O. processed and analyzed microarray and ChIP-seq data, and wrote the manuscript. M.O.L. and C.S.L. oversaw the work performed.

## Additional information

**Competing interests:** The authors declare no competing financial interests.

