## [Peer review File · Nature Communications]

Reviewers' comments:

Reviewer #1 (Remarks to the Author):

The Ms by Luo et al evaluated the role of tf GABP in the homeostasis of naive and antigen experienced T cells. Although their approach is primarily genetic (Cre mice, gene profiling, and CHIP-seq) they do also implement adoptive cell transfers in vivo to demonstrate the role of GABP in T cells upon pathogen challenge. Overall, conclusions provided by the authors that GABP plays an important role in peripheral T cell homeostasis is substantiated with the data provided. The manuscript is well written and coherently presented.

Below is a list of comments for authors to consider and address:

- 1) Wording: Authors describe GABP as being "indispensable" for T cell activation (line 80), proliferation, and survival. One can understand that none of these things are capable of occurring in the absence of GABP. The data shows this not to be true and while all three of these things are severely impacted "indispensable" is not the appropriate wording.
- 2) Discontinuities: I would like to highlight some minor discontinuities with the manuscript and ask the authors for clarification. When comparing the spleen data from figure 2 with the LN data from Supplemental 3 there is a stated difference in how they analyzed the cells. For the spleen cells were previously gated on either CD4+ or CD8+ which seems appropriate. However, in the LN the cells are gated as either CD8+ or CD4+FoxP3-. This altered gating for the CD4s in the LN does not seem to have an apparent reason and we would like the authors to either justify this alteration or redo the LN analysis without FoxP3. The next discontinuity is the use of CD8-Cre mice for part of figure 7. Up to this point and for the latter half of the figure the authors use CD4-Cre to remove GABP from all T cells. However, for this part of the figure they use CD8 and thus maintain functional CD4s. The authors are using OT-I mice in this cross which should only have CD8s with a specific TCR so it does not seem to be an experimental issue as they then transfer the cells. Yet the authors have not shown the efficacy of the CD8-Cre at removing the GABP from their CD8 T cells.
- 3) Experimental Design: Number of OT-I cells used in important co-transfer experiment (Fig 7) is superficially high. The authors should justify the use of non-physiological numbers of OT-I cells or, preferably, repeat those experiments using not more than 1000 cells transferred.
- 4) Critical data missing: No attempts were made to go beyond simple accounting experiments (Fig 7) to further document the role that GABPa plays in controlling T cell homeostasis and immunity in vivo. Are those cells that still expand upon antigen (pathogen) encounter able to function? Cytokine production, killing capacity? Are those cells capable of naive to memory transition, in other words - is memory T cell possible even in the absence of GABPa (if not, the authors should convincingly document that rejection in adoptive transfer model is not an issue). Critical phenotypic markers should be assessed in those cells at various times post infection (effector and memory time ppoints) to document and define naive to memory T cell progression in the absence of this critical tf upon infection in vivo.

Reviewer #2 (Remarks to the Author):

This paper by Luo et al., describes a novel role for GABPa in the regulation of T cell proliferation and survival in both homeostatic and activation settings.

The data showing that GABPa is required for normal T cell numbers in steady state is clear. Moreover the lack of entry into S-phase and limited cell division after activation was demonstrated in vitro and were supported by in vivo data in a number of conditions. In terms of molecular mechanism the authors show that the loss of GABPa resulted in deregulated expression of a

number of DNA replication and cell cycle genes such as *mcm3* and *mcm5* which they linked to the lack of cell cycle progression and division. They also show that GABPa binds to the regulatory regions of these genes inferring that GABPa directly regulates their expression. Overall the experiments are well presented and the data is clear. However, some of the claims are not completely supported by the data presented.

Major questions:

1. The majority of the results focus on how GABPa regulates T cells after activation but the authors also conclude that GABPa regulates T cell homeostasis by the same molecular mechanisms. This extrapolation is currently not supported by the data. Could the authors provide some molecular evidence for how GABPa regulates T cell homeostasis? If not I would suggest this paper would still be very interesting without the homeostasis data.
2. The conclusion that GABPa directly regulates the expression of *mcm3* and *5* is strong the T cell phenotype could also be caused by other gene deregulation. To support this conclusion I would like to see the effect of knocking down *mcm3* or *mcm5* in WT T cells in vitro to see if it phenocopies the loss of GABPa (lack of DNA replication/cell division/activation).

Minor questions/comments:

1. There is very little information surrounding the ChIPseq data presented in Fig S5c and Fig. 5b. What do labels on the Y-axis represent? Has the background been subtracted? The binding of GABPa to the *mcm3* and *mcm5* promoters (value of 0.5) looks to be weak given the controls in Fig. S5C.
2. Data in Figure 5e does not show large changes in *mcm5* levels that the authors refer to on page 10 of the manuscript as "attenuated". Could the authors please apply some methods of quantification to support this claim?
3. One thing that frustrated this reviewer was the inconsistency in the timepoints used between different experiments which make comparisons difficult. Why have the authors not used consistent timepoints?
4. I don't think the use of the word "severe" on page 4 to describe the peripheral T cell defect is appropriate.

Reviewer #3 (Remarks to the Author):

Luo et al. test the effect of deletion of GABPa late in thymic development (using CD4-cre) on T cell function and proliferation. The studies are generally well-designed and give clear results showing a strong proliferative defect in cko T cells, both in vitro and in vivo.

The authors stimulate their T cells with plate bound Ab against CD3 and CD28, but all the experiments depicted in figure 3, 4 and 5 were done with the addition of 100 U/mL exogenous IL-2. It is therefore remarkable that the cko cells still show such a strong proliferative defect, an important concept that is not brought out in the manuscript. But this also means that all the in vitro studies are implicating a role for GABPa downstream of the IL-2 receptor, and it is less clear how GABPa functions downstream of the TCR or TCR/CD28 (without overwhelming exogenous IL-2). The authors measure IL-2 production by ICC in a secondary restimulation experiment, but this is not particularly quantitative, and it does not give any insight into IL-2 production during the primary stimulus (also, IL2 does not appear to come up as a differentially expressed gene in the

microarray analyses). The authors need to include a detailed analysis of in vitro T cell activation and cytokine production in response to TCR/CD28 without exogenous IL-2 in order to help clarify how GABPa is functioning.

More minor issues: The authors only focus on the GABPa CHIP-seq hits that occur within a gene promoter, but they have identified nearly 6800 putative binding sites, >70% of which are not in promoters. Do any of these non-promoter sites overlap with T cell-specific open chromatin regions and/or regions enriched for H3K27ac, etc? For binding sites within potential enhancer regions, what is the ontology of nearby genes? For GABPa CHIP-qPCR validation, an important control using the specific GABPa Ab to CHIP from GABPa-cko chromatin is missing. In supplementary table 3, please clarify what 'NA' means in the Gene Symbol column. Some peaks map upstream of an 'NA', others map downstream, and others inside an NA.

Response to reviewers' comments

We would first like to thank the reviewers for evaluating our manuscript and providing constructive feedback. Due to breeding issues associated with our mouse colony, it took us longer time than anticipated to perform all the experiments to address reviewers' comments. We believe that the revised manuscript has clarified the concerns raised by the reviewers. A point-by-point response (in blue) is provided below.

Point-by-point response:

Reviewer #1 (Remarks to the Author):

The Ms by Luo et al evaluated the role of GABP in the homeostasis of naïve and antigen experienced T cells. Although their approach is primarily genetic (Cre mice, gene profiling, and CHIP-seq) they do also implement adoptive cell transfers in vivo to demonstrate the role of GABP in T cells upon pathogen challenge. Overall, conclusions provided by the authors that GABP plays an important role in peripheral T cell homeostasis is substantiated with the data provided. The manuscript is well written and coherently presented.

Below is a list of comments for authors to consider and address:

1) Wording: Authors describe GABP as being “indispensable” for T cell activation (line 80), proliferation, and survival. One can understand that none of these things are capable of occurring in the absence of GABP. The data shows this not to be true and while all three of these things are severely impacted “indispensable” is not the appropriate wording.

A: We thank the reviewer for this comment, and we have changed the word “indispensable” to “crucial” in our revised manuscript (p4).

2) Discontinuities: I would like to highlight some minor discontinuities with the manuscript and ask the authors for clarification. When comparing the spleen data from figure 2 with the LN data from Supplemental 3 there is a stated difference in how they analyzed the cells. For the spleen cells were previously gated on either CD4+ or CD8+ which seems appropriate. However, in the LN the cells are gated as either CD8+ or CD4+FoxP3-. This altered gating for the CD4s in the LN does not seem to have an apparent reason and we would like the authors to either justify this alteration or redo the LN analysis without FoxP3. The next discontinuity is the use of CD8-Cre mice for part of figure 7. Up to this point and for the latter half of the figure the authors use CD4-Cre to remove GABP from all T cells. However, for this part of the figure they use CD8 and thus maintain functional CD4s. The authors are using OT-1 mice in this cross which should only have CD8s with a specific TCR so it does not seem to be an experimental issue as they then transfer the cells. Yet the authors have not shown the efficacy of the CD8-Cre at removing the GABP from their CD8 T cells.

A: We thank the reviewer for pointing out the error in the original legend of Supplementary Fig. 3 (Supplementary Fig. 1f-g in the revised version). In fact, we gated on CD4⁺, not CD4⁺Foxp3⁻ cells in the lymph nodes. We have corrected it in the revised manuscript.

*Regarding the OT-1 mice used in *Listeria monocytogenes* infection, we chose CD8^{Cre} rather than CD4^{Cre} for more efficient deletion of GABP α . Our previous experience with the CD4^{Cre} OT-1*

model revealed incomplete depletion of floxed alleles in CD8⁺ T cells (data not shown), which is caused by earlier TCR expression and the bypassing of the DP stage for some OT-1 TCR⁺ cells.

The efficacy of CD8^{Cre} at removing GABP protein from CD8⁺ T cells is shown by western blotting experiments shown below.

Revision Figure 1:

Revision Figure 1: Efficient GABPα deletion in the CD8⁺ T cells from CD8^{Cre}GABP^{ff} mice. Immunoblotting analysis of GABPα in purified CD8⁺ T cells from the spleen and lymph nodes of *Gabpa^{ff}* (WT) and *CD8^{Cre}GABP^{ff}* (KO) mice.

3) *Experimental Design: Number of OT-1 cells used in important co-transfer experiment (Fig 7) is superficially high. The authors should justify the use of non-physiological numbers of OT-1 cells or, preferably, repeat those experiments using not more than 1000 cells transferred.*

A: We decided to transfer relatively high numbers of OT-1 cells in order to analyze the cells at an early time point after infection (day 3). In addition, similar phenotype was observed in the polyclonal system (*CD4^{Cre}Gabpa^{ff}*, Fig. 7f-g), suggesting that the defects of GABP-knockout T cells in response to antigen stimulation were not caused by the high frequency of antigen-specific T cells.

4) *Critical data missing: No attempts were made to go beyond simple accounting experiments (Fig 7) to further document the role that GABPα plays in controlling T cell homeostasis and immunity in vivo. Are those cells that still expand upon antigen (pathogen) encounter able to function? Cytokine production, killing capacity? Are those cells capable of naïve to memory transition, in other words - is memory T cell possible even in the absence of GABPα (if not, the authors should convincingly document that rejection in adoptive transfer model is not an issue). Critical phenotypic markers should be assessed in those cells at various times post infection (effector and memory time points) to document and define naïve to memory T cell progression in the absence of this critical tf upon infection in vivo.*

A: We analyzed the very low numbers of antigen-specific cells in *CD4^{Cre}Gabp^{ff}* mice seven days post LM-OVA infection (Supplemental Fig. 8). We found that GABPα-replete and -depleted mice contained similar fractions of IL-7Rα^{hi}KLRG1^{lo} memory-precursor effector T cells (MPECs) and IL-7Rα^{lo}KLRG1^{hi} short-lived effector T cells (SLECs) among the Kb-ova⁺ cells. Moreover, GABPα-deficient antigen-specific CD8 T cells expressed comparable levels of granzyme B, and similar or slightly compromised levels of IFN-γ as their wild-type counterparts. These data imply that the cells in *CD4^{Cre}Gabp^{ff}* mice that expanded upon pathogen encounter have similar memory potential, killing capacity and cytokine production as their wild-type counterparts. More detailed characterization of the role of GABPα in memory responses will need an experimental

system that deletes GABP α in antigen-experienced T cells, which is beyond the scope of this manuscript.

Reviewer #2 (Remarks to the Author):

This paper by Luo et al., describes a novel role for GABP α in the regulation of T cell proliferation and survival in both homeostatic and activation settings.

The data showing that GABP α is required for normal T cell numbers in steady state is clear. Moreover the lack of entry into S-phase and limited cell division after activation was demonstrated in vitro and were supported by in vivo data in a number of conditions. In terms of molecular mechanism the authors show that the loss of GABP α resulted in deregulated expression of a number of DNA replication and cell cycle genes such as mcm3 and mcm5 which they linked to the lack of cell cycle progression and division. They also show that GABP α binds to the regulatory regions of these genes inferring that GABP α directly regulates their expression. Overall the experiments are well presented and the data is clear. However, some of the claims are not completely supported by the data presented.

Major questions:

1. The majority of the results focus on how GABP α regulates T cells after activation but the authors also conclude that GABP α regulates T cell homeostasis by the same molecular mechanisms. This extrapolation is currently not supported by the data. Could the authors provide some molecular evidence for how GABP α regulates T cell homeostasis? If not I would suggest this paper would still be very interesting without the homeostasis data.

A: We thank the reviewer for this comment. In addition to the defects shown in response to antigen stimulation (Fig. 3, 5 and 7), we also observed diminished homeostatic proliferation in GABP-deficient T cells (Fig. 6). Moreover, GABP-deficient mice had about 2 fold reduction of splenic and lymph nodes TCR⁺ cells compared to littermate controls (Fig. 2a-c, and Supplementary Fig. 1a-c). We don't think that the defects in antigen-experienced T cells alone are sufficient to explain the phenotype.

We have also performed additional in vitro culture experiments to assess the role of GABP α in T cell homeostasis. We sorted naive T cells from CD4^{Cre}Gabp^{fl/fl} mice and littermate controls, and cultured them in the presence of IL-7. We found that although the cells were not activated, loss of GABP α resulted in compromised T cell survival (Supplementary Fig. 2).

We don't yet have a complete understanding regarding the molecular mechanisms of GABP α -dependent homeostatic regulation. However, we speculate that GABP α controls similar programs under homeostatic conditions and upon antigen stimulation, because the majority of GABP α direct target genes were differentially expressed at both 0h and 18h time points between WT and KO cells, with the difference more exaggerated at 18h (Fig. 4c, and Supplementary Table 4).

2. The conclusion that GABP α directly regulates the expression of mcm3 and 5 is strong the T cell phenotype could also be caused by other gene deregulation. To support this conclusion I would like to see the effect of knocking down mcm3 or mcm5 in WT T cells in vitro to see if it phenocopies the loss of GABP α (lack of DNA replication/cell division/activation).

A: We thank the reviewer for suggesting this experiment. We knocked down Mcm3 in WT T cells using shRNA, and found that the incorporation of EdU in response to anti-CD3 and anti-CD28 stimulation was reduced about 30% in CD8⁺ T cells compared with the scramble shRNA control (Supplementary Fig. 7), suggesting that defective proliferation of GABP α -deficient T cells is caused at least in part by reduced expression of Mcm proteins.

Minor questions/comments:

1. There is very little information surrounding the ChIPseq data presented in Fig S5c and Fig. 5b. What do labels on the Y-axis represent? Has the background been subtracted? The binding of GABP α to the mcm3 and mcm5 promoters (value of 0.5) looks to be weak given the controls in Fig. S5C.

A: The Y-axis in Fig. S4 (Supplementary Fig. 5 in the revised version) and Fig. 5b represent read per million (RPM). The background has been subtracted in the revised figures. Regarding the binding strength of GABP α , we picked very strong peaks in Fig. S5 as representative positive controls. Although the binding of GABP α to the Mcm3 and Mcm5 promoters was weaker compared with the controls shown, it's statistically significant, and has been verified by ChIP-qPCR experiments (Fig. 5d).

2. Data in Figure 5e does not show large changes in mcm5 levels that the authors refer to on page 10 of the manuscript as "attenuated". Could the authors please apply some methods of quantification to support this claim?

A: We thank the referee for this comment. We have added quantifications in the revised Fig. 5. The relative amounts of proteins were normalized to the 0-hour time point of each individual group.

3. One thing that frustrated this reviewer was the inconsistency in the timepoints used between different experiments which make comparisons difficult. Why have the authors not used consistent time points?

A: We chose multiple time points (0h, 12h, 36h and 72h) in Fig. 3 in order to gain a more comprehensive understanding of how GABP α -deficient T cells respond to antigen stimulation. Data from Fig. 3 showed that most drastic phenotypical changes occurred between 12h and 36h time points, which prompted us to pick a time point in between (18h) to perform the transcriptome analysis, in order to explore the primary cause of the phenotypes observed. For the cell cycle experiments shown in Fig. 5, we followed established protocols from literature and used the same time points to assess EdU incorporation and expression changes of cell cycle machineries.

4. I don't think the use of the word "severe" on page 4 to describe the peripheral T cell defect is appropriate.

A: We removed the word "severe" in the text.

Reviewer #3 (Remarks to the Author):

Luo et al. test the effect of deletion of GABP α late in thymic development (using CD4-cre) on T

cell function and proliferation. The studies are generally well-designed and give clear results showing a strong proliferative defect in cko T cells, both in vitro and in vivo.

The authors stimulate their T cells with plate bound Ab against CD3 and CD28, but all the experiments depicted in figure 3, 4 and 5 were done with the addition of 100 U/mL exogenous IL-2. It is therefore remarkable that the cko cells still show such a strong proliferative defect, an important concept that is not brought out in the manuscript. But this also means that all the in vitro studies are implicating a role for GABPa downstream of the IL-2 receptor, and it is less clear how GABPa functions downstream of the TCR or TCR/CD28 (without overwhelming exogenous IL-2). The authors measure IL-2 production by ICC in a secondary restimulation experiment, but this is not particularly quantitative, and it does not give any insight into IL-2 production during the primary stimulus (also, IL2 does not appear to come up as a differentially expressed gene in the microarray analyses). The authors need to include a detailed analysis of in vitro T cell activation and cytokine production in response to TCR/CD28 without exogenous IL-2 in order to help clarify how GABPa is functioning.

A: We thank the reviewer for suggesting this experiment. We performed a detailed analysis of *in vitro* T cell activation and cytokine production in response to anti-CD3 and anti-CD28 stimulation in the absence of exogenous IL-2. Similar defects in cell activation, proliferation and survival in the GABP-deficient T cells were observed (Revision Fig. 2), implying that the GABP-mediated regulation is independent of IL-2 receptor signaling. We agree with the reviewer that ICCS is not a particularly quantitative method in determining IL-2 production. But here we use IL-2 production as a readout for T cell activation, since it's a salient characteristic of activated T cells.

Revision Figure 2

Revision Figure 2: GABP α is required for T cell activation, proliferation and survival in response to antigen stimulation *in vitro*. Naive (CD62L^{hi} CD44^{lo}) CD4⁺ or CD8⁺ T cells from *Gabpa*^{f/f} (WT) and *CD4^{Cre}Gabpa*^{f/f} (KO) mice were purified by flow cytometric sorting, and were subjected to anti-CD3 and CD28 stimulation in the presence (**a-d**) or absence (**e-h**) of IL-2. Representative plots from CD8⁺ T cell culture were shown. **a, e**, Analysis of cell size (FSC) and activation markers, CD69 and CD25, at 0h, 12h and 36h post stimulation. **b, f**, 72h after cell culture, WT and KO cells were restimulated with PMA and ionomycin for 4h and analyzed for the expression of IL-2 by intracellular cytokine staining. **c, g**, WT and KO cells were labeled with the cytosolic dye CFSE, and cell division was assessed by the dilution of CFSE. Grey shaded line shows CFSE level of undivided cells. **d, h**, Cell death was assessed with Annexin V and 7-AAD staining.

More minor issues: The authors only focus on the GABP α ChIP-seq hits that occur within a gene promoter, but they have identified nearly 6800 putative binding sites, >70% of which are not in promoters. Do any of these non-promoter sites overlap with T cell-specific open chromatin regions and/or regions enriched for H3K27ac, etc? For binding sites within potential enhancer regions, what is the ontology of nearby genes? For GABP α ChIP-qPCR validation, an important control using the specific GABP α Ab to ChIP from GABP α -cko chromatin is missing. In

supplementary table 3, please clarify what 'NA' means in the Gene Symbol column. Some peaks map upstream of an 'NA', others map downstream, and others inside an NA.

A: We thank the reviewer for this comment. To determine the overlap between GABP α binding sites and open chromatin regions, we used the H3K27ac ChIP sequencing data from naïve CD4 T cells published by Rudensky lab for the analysis (GSM1893205, GSM1893206, GSM1893219, GSM1893220). We identified 23310 H3K27ac peaks from the 4 datasets listed above, and then analyzed the overlap between GABP α peaks in non-promoter regions (≥ 1 kb upstream or downstream of TSS) and the H3K27ac peaks. There are a total of 3462 GABP α binding peaks in the non-promoter regions. Out of these binding sites:

- 1) 316 GABP α binding peaks overlap with H3K27ac peaks, if the cutoff is that the GABP α peak and the H3K27ac peak have ≥ 1 bp overlap;
- 2) 555 GABP α binding peaks overlap with H3K27ac peaks, if the cutoff is that the gap between the GABP α peak and the H3K27ac peak is ≤ 1000 bp.

We agree with the reviewer that performing ChIP experiments with GABP α antibody in GABP α -knockout cells can be used to show the specificity of our GABP α ChIP. However, *CD4^{Cre}GABP^{f/f}* mice harbor significantly reduced number of T cells in the periphery (Fig. 2 and Supplementary Fig.1), making it highly technically challenging to perform ChIP experiments from these mice. For the Mcm3 and Mcm5 GABP binding sites shown in Fig.5d, the GABP α enrichment is at least 10 fold greater than the IgG enrichment, whereas the GABP α enrichment of the Hprt1 locus (negative control) is comparable as the IgG enrichment.

Regarding the "NA" in the Gene Symbol column of Supplementary Table 3, we used ensembl IDs to annotate genes. Although some genes have ensembl IDs, they don't have gene symbol annotation.

REVIEWERS' COMMENTS:

Reviewer #1 (Remarks to the Author):

none

Reviewer #2 (Remarks to the Author):

The authors have provided adequate answers to my concerns.

Reviewer #3 (Remarks to the Author):

Please include in vitro activation data in the manuscript either in the main figure or as supplemental data

Response to reviewers' comments

We would like to thank the reviewers again for taking the time to evaluate our manuscript. A point-by-point response (in blue) is provided below.

Point-by-point response:

Reviewer #1 (Remarks to the Author):

none

Reviewer #2 (Remarks to the Author):

The authors have provided adequate answers to my concerns.

Reviewer #3 (Remarks to the Author):

Please include in vitro activation data in the manuscript either in the main figure or as supplemental data.

We thank the reviewer for this comment. The in vitro activation data has been added in the manuscript as Supplementary Figure 3.